# A Combination of Dilated Self-Attention Capsule Networks and Bidirectional Long- and Short-Term Memory Networks for Vibration Signal Denoising

**Youming Wang \*, Gongqing Cao and Jiali Han**

School of Automation, Xi'an Key Laboratory of Advanced Control and Intelligent Process, Xi'an University of Posts and Telecommunications, Xi'an 710121, China
* Correspondence: xautroland@126.com; Tel.: +86-29-8816-6341

**Abstract:** As scalar neurons of traditional neural networks promote dimension reduction caused by pooling, it is a difficult task to extract the high-dimensional spatial features and long-term correlation of pure signals from the noisy vibration signal. To address the above issues, a vibration signal denoising method based on the combination of a dilated self-attention capsule network and bidirectional long short memory network (DACapsNet–BiLSTM) is proposed to extract high-dimensional spatial features and learn long-term correlations between two adjacent time steps. An improved self-attention module with spatial feature extraction ability was constructed based on the random distribution of noise, which is embedded into the capsule network for the extracted spatial features and denoising. The dilated convolution is integrated into the improved capsule network to expand the receptive field to obtain the spatial features of the vibration signal. The output of the capsule network was used as the input of the bidirectional long-term and short-term memory network to obtain the timing characteristics of the vibration signal. Numerical experiments demonstrated that DACapsNet–BiLSTM performs better than other signal denoising methods, in terms of signal-to-noise ratio, mean square error, and mean absolute error metrics.

**Keywords:** capsule networks; bi-directional long- and short-term memory networks; vibration signal; dilated convolution; denoising; self-attention





## 1. Introduction

Rolling bearing operating conditions are the main factor affecting the overall operating condition of rotating machinery [1]. The fault diagnosis method based on the rolling bearing is feature extraction of vibration signals to determine the fault status. However, due to the inherent physical limitations of various acquisition devices, the collected vibration signals contain many interfering signals, which have a negative impact on the subsequent fault diagnosis [2]. Therefore, it is necessary to suppress noise to optimize the signal-to-noise ratio and provide accurate feature extraction of the vibration signal.

In the past two decades, researchers have continuously thrived to investigate efficient denoising algorithms, including spatial domain filtering, transform-domain thresholding, sparse representation, etc., and aimed to restore reasonable estimates from different vibration signals while preserving fault features. Traditional vibration signal denoising methods include wavelet thresholding denoising [3–5], empirical mode decomposition (EMD) [6,7], singular value decomposition (SVD) denoising [8], local mean decomposition (LMD) [9], filter denoising [10,11], and variable component modal decomposition (VMD), etc. Smith [12] proposed the LMD algorithm for the denoising of EEG perception data, which used frequency and energy structure to analyze a wide variety of natural signals. Dragomiretskiy et al. [13] proposed a VMD algorithm for the denoising of the artificial and real signals by decomposing a signal into an ensemble of band-limited intrinsic mode functions. The SVD is also a denoising algorithm for solving linear least squares problems

with rank deficiencies. Among of the above methods, the performance of signal processing is related to parameter selection, which usually depends on the experience of the engineer.

In recent years, methods based on deep learning, such as deep neural networks (DNNs) [14], denoising auto-encoders (DAEs) [15], convolutional neural networks (CNNs) [16], and recurrent neural networks (RNNs), have developed rapidly in various bailiwicks such as audio, denoise image, natural language processing, and fault diagnosis. Compared with traditional denoising methods, signal denoising methods based on deep learning can learn the nonlinear correlation among the noisy and the original signal. During the last several years, convolutional neural networks (CNNs) have been significantly applied in the field of denoising. Convolutional kernels are used to map the input features in CNNs, which integrate global features through the learning of local features. Several improvisations for vibration signal denoising based on CNN methods have been developed afterward to accelerate filter implementation and to improvise the qualitative and quantitative results. Jain et al. [17] used CNN for the first time for denoising and demonstrated that convolutional neural networks can directly learn the end-to-end nonlinear mapping from the underlying image-to-image subjected to Gaussian noise with good results. Lou et al. [18] proposed a new denoising autoencoder model based on simplified convolution to extract the superior features to eliminate noise in a chaotic signal. Fan et al. [19] constructed a vibration signal denoising model based on the ResNet network to eliminate noise from measured vibration data.

However, as a data-driven approach, CNN-based methods require a stronger long-range correlation learning capability in order to be able to completely separate the noise from the original signal. In traditional CNN, stacked convolutional layers are used to increase the perceptual field to improve global feature extraction, which leads to deeper network layers and reduced convergence. This further leads to phenomena including gradient disappearance or explosion. Dilated convolution was proposed to enlarge the receptive field with fewer parameters to solve above problem [20], which was used for fault diagnosis to extract rich fault features. Kumar et al. [21] constructed a dilated convolutional network for fault diagnosis. Chu et al. [22] constructed a multi-scale network based on dilation rates and an attention mechanism for fault detection. Wu et al. [23] proposed a convolutional neural network with a novel loss function to extract features from seismic data. However, the distortion of test data and the random distribution of noise will weaken the feature extraction ability of CNN and lead to performance degradation. An attention mechanism is introduced to focus on different feature information in the influence of complex data, based on the feature of adaptive weight allocation. Wang et al. [24] proposed a convolutional neural network with an attention mechanism to deal with vibration signals in complex environment. Fu et al. [25] constructed a CNN and long–short time hybrid neural network based on a self-attentive mechanism for the prediction of temporal signals. Zhong et al. [26] proposed an improved lightweight convolutional neural network based on transfer learning for bearing fault detection. The pooling layers provide the prior probability of translational invariance for the model; however, they ignore the specific spatial information in CNNs. The azimuthal features cannot be fully extracted by the CNN. In addition, CNNs learn more spatial features by stacking convolutional layers to expand the perceptual domain, which ignores the relative position information between features. Therefore, CNN classification performance decreases in the face of complex data classification containing relative location information. To overcome the above shortcomings, Sabour et al. [27] introduced the capsule network (CapsNet), where multidimensional vector neurons are used as capsules to encode the relative location features of objects, and dynamic routing algorithms are used to pass important information between the capsules [28]. CapsNet uses a capsule layer instead of a pooling layer to efficiently retain valuable information and positional relationships in the signal without the addition of additional parameters. Scholars have tried to implement the CapsNet model in various applications, such as image recognition [29–31], traffic forecasting, fault diagnosis, etc. [32,33]. Liu et al. [34] applied the capsule network to extract the unbalanced fault feature of rolling bearings to enhance the classification accuracy of

networks. Long et al. [35] proposed a deep learning model for diagnosing robot faults by fusing the feature extraction capability of CNNs and the learning capability of CapsNet for spatial location.

Although capsule networks make full use of stacked convolutional layers to capture the spatial correlation of sequential data, they ignore the dependency features between local information [36]. Recurrent neural networks (RNNs) perform well against sequential data, but the gradient disappearance during backpropagation deteriorates their performance; therefore, a "gate" structure is added to the long and short term memory (LSTM) to address the above issues [37]. As a further development of the LSTM model, the bidirectional-LSTM (BiLSTM) model captures finer-grained correlations between data by accessing both sequence directions [38]. Cui et al. [39] constructed a multi-channel speech enhancement algorithm based on multiple targets, which uses BiLSTM networks to deal with noise and reverberation problems. Shi et al. [40] introduced a BiLSTM-based network model to classify faults by extracting spatial and temporal features from the complex position–direction information of gearboxes.

Aiming at the above problems, we noted that the original CapsNet model ignores the temporal features of the time series data for vibration signals and the temporal and spatial features of noise. Therefore, we aimed to design a combined DACapsNet–BiLSTM model to capture the discriminative fault features, which can maintain the advantages of spatial information in CapsNet and bidirectional temporal information in LSTM, as well as remove the noise information in the vibration signal. In comparison to other single structure models, the proposed model embeds dilated convolution in the capsule network part to enhance the spatial extraction capability of the network, and BiLSTM is applied to deal with temporal features in time-series data. As noise has the property of being randomly distributed in space, we propose a novel location self-attentive mechanism to remove the noise. The main contributions of this paper include as follows.

(1) A hybrid neural network for signal denoising is proposed, which takes the original time signal as input to denoise the vibration signal.

(2) The one-dimensional vibration signal is transformed into a two-dimensional image as the input to the DACapsNet–BiLSTM model. The dilated convolution is integrated into the network model to enlarge the receptive field of the network and strengthen the feature extraction ability of the proposed model. The improved self-attention mechanism is embedded into the capsule network, in which the location focus mechanism is introduced to enhance the ability to extract the temporal characteristics of vibration signals and suppress the influence of random distribution of noise.

(3) BiLSTM networks were used to extract the temporal features of vibration signals to enhance the weakness of capsule neural networks in modeling long-term dependencies.

The remainder of this paper is structured as follows. Section 2 introduces the basic principles including signal denoising, dilated convolution, self-attention, CapsNet, and BiLSTM. Section 3 presents the proposed vibration signal denoising model based on the DACapsNet–BiLSTM network. In Section 4, a comparative experiment is conducted to verify the validity of the DA model, and the article is concluded with the conclusion in Section 5.

## 2. Preliminaries

### 2.1. Signal Denoising

The common form of a denoising problem can be expressed by

$$s(n) = x(n) + \delta e(n) \tag{1}$$

where $s(n)$ is the real signal, $x(n)$ is the clean signal, $e(n)$ is the noise and $\delta$ is the noise level. The goal of denoising is to eliminate the noise from the real signal.

It is assumed that noises do not interfere with each other. The total noise conforms to the probability distribution of white Gaussian noise based on the central limit theory and follows a normal distribution.

$$e(n) \sim N\left(0, \sigma^2\right) \tag{2}$$

where $\sigma^2$ is the power of the noise, $N$ is the normal distribution. These noises are randomly generated and spuriously distributed, which can disrupt the inherent features of the signal.

### 2.2. Dilated Convolution

In general, orientation features are necessary when it regards to signal processing. Dilated convolution is used to increase the field of perception and to collect large amounts of spatio-temporal data for accurate recognition, which can improve the field of perception by generating zeros between pixels in the convolution kernel that can be applied to learn the global distribution of feature information without sacrificing resolution. The dilated convolution kernel with an expansion rate of 2 has the same perceptual field as the convolution kernel, but with only 9 parameters, which is 36% of the number of convolution parameters. This enrichment of the output features of each convolution by increasing the perceptual field without decreasing the accuracy allows the application of dilated convolution in domains with long-term dependencies.

### 2.3. Self-Attention

A self-attention structure is seen in classical architecture, which calculates the correlation matrix between different features to extract the long-distance correlation to enhance the overall the performance of the network as a whole as shown in Figure 1. Suppose $X \in R^{n \times d}$ is a sequence sample of input features. The $n$ is the number of input samples, and $d$ is the latitude of a single sample. The definition of the correlation output of self-attention is expressed by

$$Output(K, Q, V) = softmax\left(\frac{QK^T}{\sqrt{d_k}}\right)V \tag{3}$$

where $K$, $Q$ and $V$ are different transformations of input $X \in R^{n \times d}$, and the long-range correlation is captured though the correlation matrix obtained by multiplying $K$, $Q$, and $V$, and $d_k$ is a scaling factor, respectively. $T$ is the transposition of the matrix $K$.

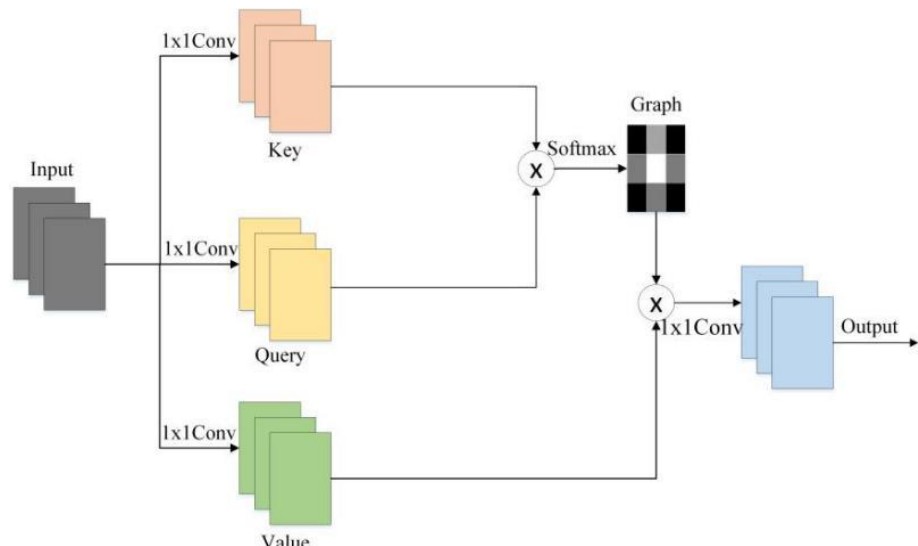

**Figure 1.** The architecture of self-attention.

### 2.4. Spatial Features Captured by CapsNet

Capsule networks can maintain the relative position information in the data through a set of neurons, and can preserve the key features of the vibration signal. Capsule

neural networks provide an abstract representation of describing multidimensional features through vectors, where the length and direction of the vectors represent the feature inherent probability information and feature location information, respectively, and the features are passed through a dynamic routing algorithm by the clustering algorithm. The architecture of capsule network structure is described in Figure 2.

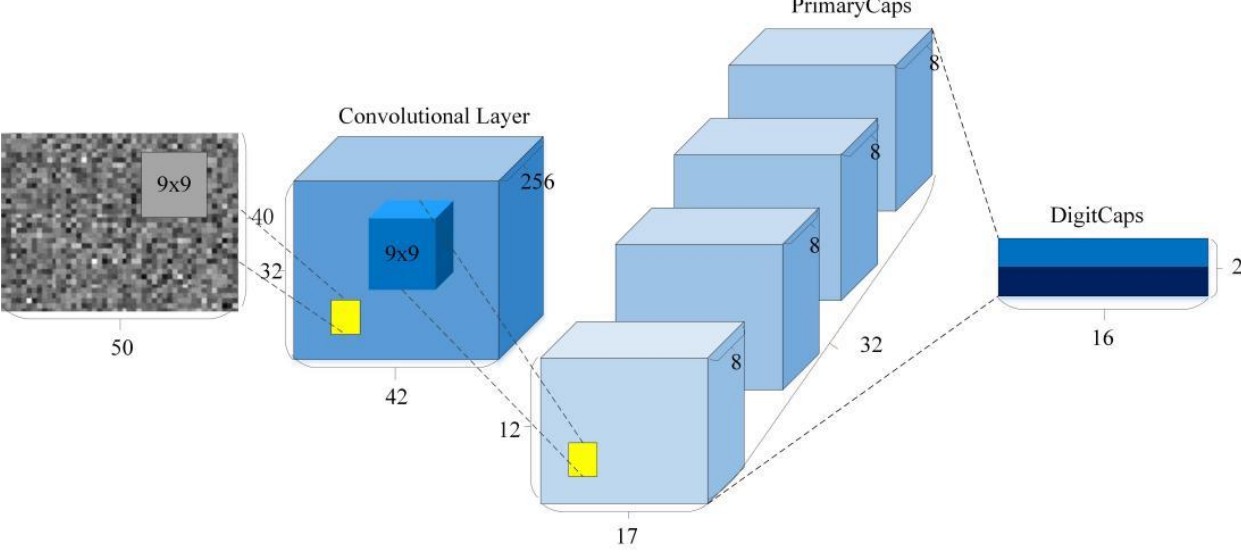

**Figure 2.** The architecture of CapsNet.

As shown in Figure 2, the local areas of the input signal are convolved using the convolutional kernel, and the sliding operation of the convolutional kernel is used to convolve the input data of the whole upper layer. The primary capsule layer and the digital capsule layer encode the input feature information to encapsulate the feature information and reduce the parameters without losing the feature information. The activation function is used to map the one-dimensional feature information to the spatial dimension, which can be described by

$$X_j^l = f\left(\sum_{i=1}^{N} x_i^{l-1} \cdot k_{ij}^l + b_j^l\right) \tag{4}$$

where $N$ is the number of convolutional kernels in the $l-1$th layer, $X_j^l$ and $x_i^{l-1}$ are the output and input of the convolutional kernels, $k$ and $b$ are the respective kernels and bias, and $f$ is the nonlinear activation function. The ReLU function is selected into the activation function of the convolution output.

The main capsule layer is the first capsule layer where the scalar-valued feature extractor is replaced with a vector-valued capsule. The output of the main capsule is expressed by

$$u^{l(i,j)} = f_s\begin{pmatrix} f_a\left(z_1^{l(i,j)}\right) \\ f_a\left(z_2^{l(i,j)}\right) \\ \vdots \\ f_a\left(z_m^{l(i,j)}\right) \end{pmatrix} = f_s\begin{pmatrix} f_a\left(K_{1i}^l \times x^{l(r^j)}\right) \\ f_a\left(K_{2i}^l \times x^{l(r^j)}\right) \\ \vdots \\ f_a\left(K_{mi}^l \times x^{l(r^j)}\right) \end{pmatrix} \tag{5}$$

where $u^{l(i,j)}$ represents the primary capsule, $f_a\left(z_m^{l(i,j)}\right)$ is the output of the convolutional layer after activation, and $f_s$ is the squeeze function. The squashed nonlinear function is used as the activation function in the capsule neural network to secure the output vector length between 0 and 1.

The dynamic routing algorithms are employed to perform selective connections between the primary capsule layer and the digital capsule layer, as shown in Figure 3. Forecasts for high-level capsules are expressed by

$$u_{j|i} = W_{ij}u_i \tag{6}$$

where $u_i$ is $i$-th input capsule, $W_{ij}$ is the weighting matrix, and $u_{j|i}$ is the forecast vector. The input vectors of advanced capsules are the weighted sum of all their prediction vectors.

$$S_j = c_{ij}u_{j|i} \tag{7}$$

where $c_{ij}$ is the coupling factor and satisfies the equation for $\sum_i c_{ij} = 1$. The coupling coefficients $c_{ij}$ can be described by

$$c_{ij} = \frac{e^{b_{ij}}}{\sum_j e^{b_{ij}}} \tag{8}$$

where $b_{ij}$ is the log prior probability the aggregation of capsule $i$ and the capsule $j$, and $b_{ij}$ is updated as

$$b_{ij} = b_{ij} + W_{ij}u_i v_j \tag{9}$$

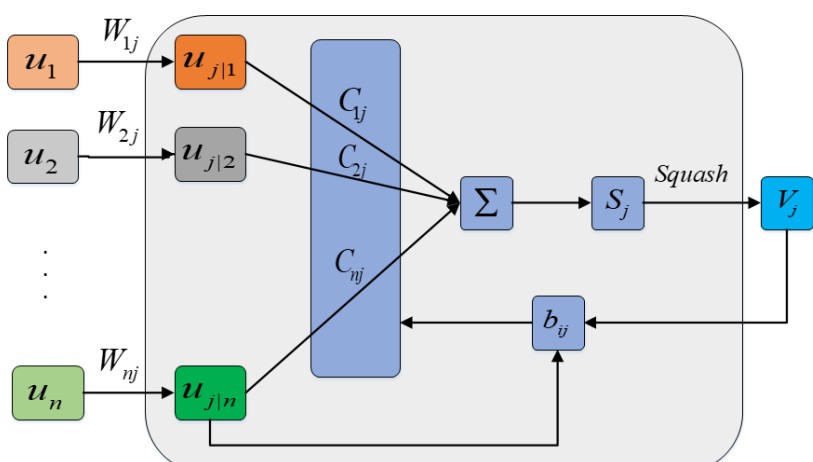

**Figure 3.** Dynamic routing algorithm.

The vector of the capsule is output through a "squashed" activation function by

$$V_j = Squash(S_j) = \frac{\|S_j\|^2}{1 + \|S_j\|^2} \frac{S_j}{\|S_j\|} \tag{10}$$

where $V_j$ is the output of high-level capsule, $\frac{S_j}{\|S_j\|}$ is the direction of the vector, and $\frac{\|S_j\|^2}{1+\|S_j\|^2}$ indicates the scaling factor.

### 2.5. BiLSTM

BiLSTM and LSTM have recurrent neural network (RNN) architecture for processing sequential data, as shown in Figures 4 and 5. The BiLSTM is one of the improvements of LSTM that stores historical information while checking the relationship between two directions of data. LSTM flexibly solves long-term related issues in the form of a gate structure. Three control units are introduced into the LSTM model, including input gates, output gates, and forgetting gates, as well as a memory unit, as shown in Figure 4. The gate structure ensures the selective passage of information. The input gate is used to choose

the important input feature that need to be stored, the forgetting gate is used to drop the non-essential feature, and the output gate is used to output the selected feature.

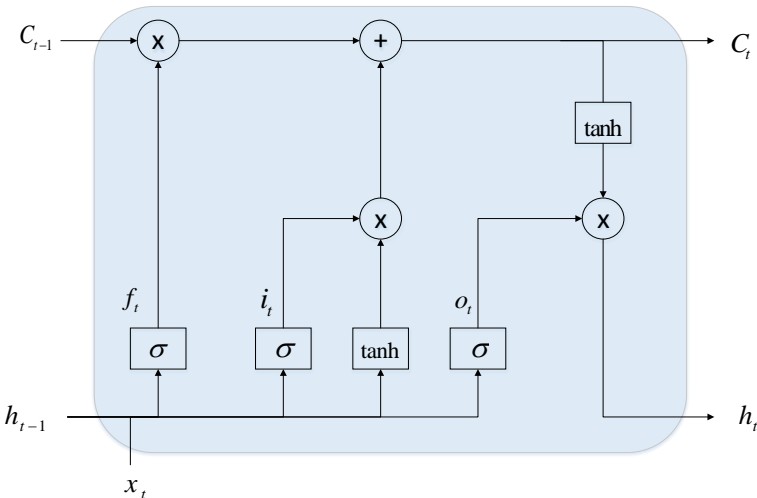

**Figure 4.** Structure of the LSTM network.

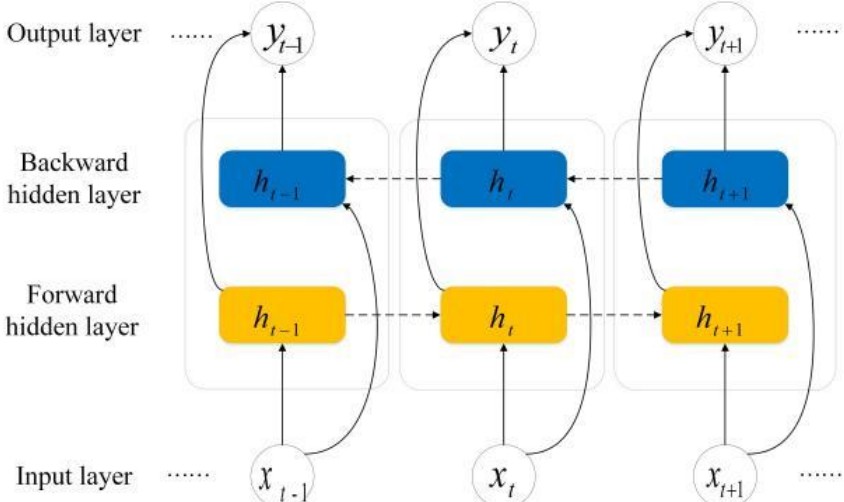

**Figure 5.** Structure of the BiLSTM network.

The architecture of BiLSTM is shown in Figure 5. The output of BiLSTM is a series of output values of forward information propagation and output values of backward information propagation, which can consider the correlation between the nodes before and after the data.

## 3. DACapsNet–BiLSTM Network for Signal Denoising

In this section, an end-to-end intelligent denoising method (DACapsNet–BiLSTM) is proposed to extract the spatial and temporal feature in the vibration signal by improving the deep learning architecture and output the denoised vibration signal. DACapsNet–BiLSTM can extract spatial features and temporal features of vibration signals by cascading improved capsule networks and BiLSTM networks. The improved capsule network is embedded with a self-attention mechanism with the spatial attention to enhance the denoising capability of the network, which is used as an input to the BiLSTM to extract timing feature.

### 3.1. Data Preprocessing

Two-dimensional grayscale images contain more feature information. The one-dimensional vibration signal is transformed into a two-dimensional gray scale image and fed into

the capsule network for feature extraction. The vibration signal possesses a periodicity, which implies that the signal state of the current moment associated with the state of the nearby moment is also associated with the nearby period. The two-dimensional matrix representation can help the network to learn the periodic characteristics of the vibration signal sufficiently, as shown in Figure 6.

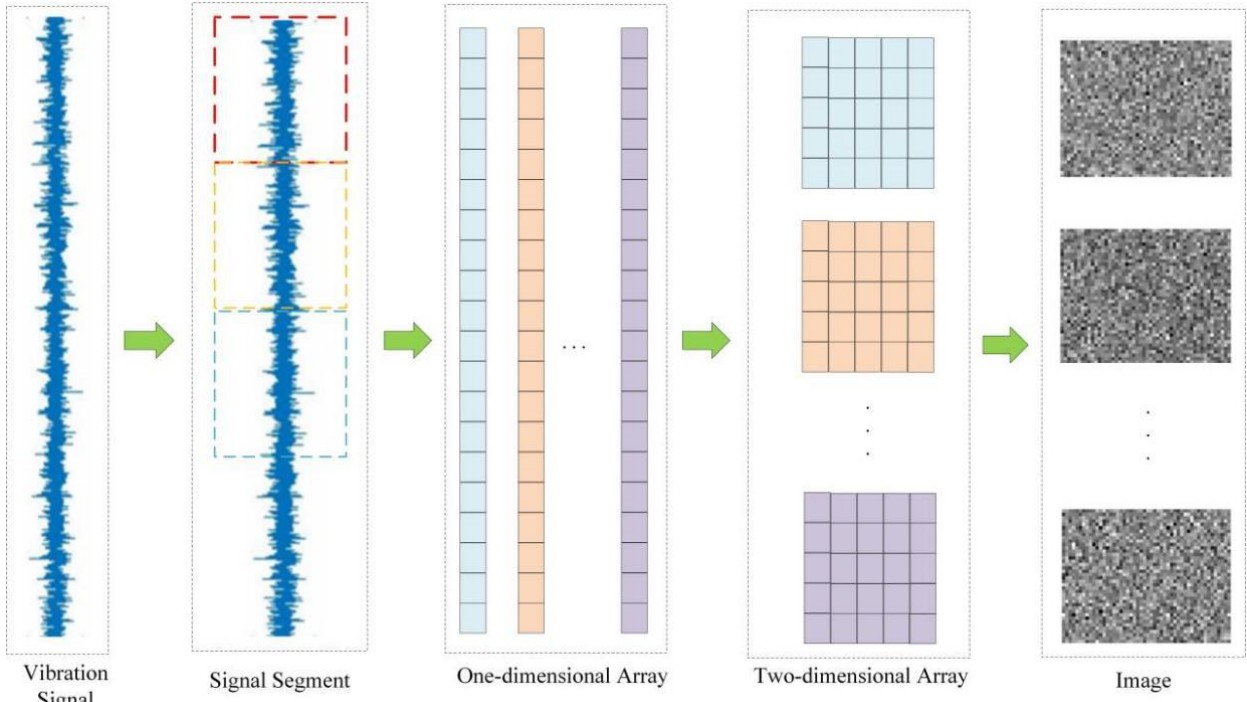

**Figure 6.** Data processing flow chart.

The vibration signal is segmented using a sliding window, and the segmented vibration signal data are sequentially used as the rows of the vibration matrix. The mathematical function can be described by

$$I = \begin{bmatrix} x(t) \cdots x(t+n-1) \\ \vdots \\ x(t+(m-1)n) \cdots x(t+mn-1) \end{bmatrix} \tag{11}$$

where $I$ is the signal image and $x(t)$ is the vibration signal data of time $t$.

### 3.2. DACapsNet–BiLSTM Model

3.2.1. Framework

As shown in Figure 7, a vibration signal denoising method based on the DACapsNet–BiLSTM network for bearings is proposed, which is composed of an improved CapsNet module and BiLSTM module. The single layer convolution in the traditional capsule network is replaced by the dilated convolution to increase the receptive field and extract the shallow features of the vibration image. Due to the complexity of input data and the random distribution of noise, the self-attention of location attention mechanism is embedded to focus on spatial features. BiLSTM is introduced to focus on the periodicity and long correlation of the time series vibration signals.

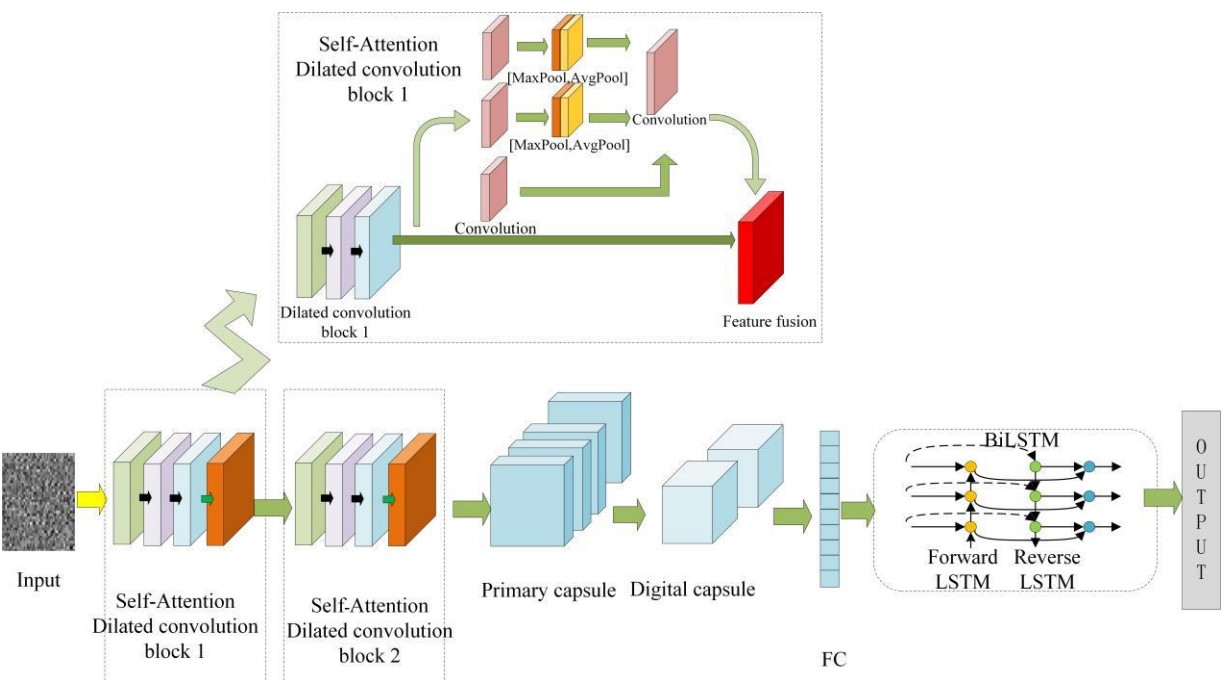

**Figure 7.** DACapsNet–BiLSTM model.

### 3.2.2. Proposed Model
Dilated Self-Attention Convolution Network

Dilated convolution was used instead of ordinary convolution to expand the receptive field, which is helpful for capsule network to extract the overall characteristics of vibration signals. Dilated convolution has fewer parameters and decreases the training burden of the model compared with traditional convolution. Under the same conditions, the receptive field can be increased to prevent the loss of feature due to the use of the lower sampling layer. The calculation formula of equivalent convolution kernel size in dilated convolution can be described by

$$K = (r - 1)(k - 1) + k \tag{12}$$

where $k$ is the convolution kernel size, $r$ is the dilated rate, and $K$ is the equivalent convolution kernel size. The dilated ratio is changed to obtain a larger receptive field, and zero filling was used to keep the size of the feature image after dilated convolution unchanged, which allows the convolution kernel to expand the receptive field without merging operations and information loss.

The expansion of the receptive field can be achieved by stacking expanded convolutional layers and setting different expansion rates as the depths increases. The features of the signal image output by convolution were extracted by cascading three-layer dilated convolution. As shown in Figure 8, where $k$ is the convolution kernel size and $r$ is the dilated ratio.

The self-attention layer was added to enhance the contribution of features in the network at important moments to the current moment, where the location attention mechanism was embedded in the self-attention layer to eliminate noise from random distributions, as shown in Figure 9. The positional self-attention mechanism assigns different weights to different local learning of the input sequence and learns the self-attention weights at different times by the output key feature matrix and value feature matrix. The two different pooling operations were used to aggregate the channel information of the two feature maps. The original Key feature matrix and Value feature matrix were weighted separately to

output feature maps through self-attentiveness to improve the spatial feature extraction capability of the model; the formula can be described by

$$
\begin{aligned}
c &= \quad f^{conv} \cdot f^{Attention} \\
&= \quad f^{conv}(softmax(Key', Query') \cdot Value) \\
&= \quad f^{conv}(softmax(Key \cdot (f^{conv}[MaxPool(Key); AvgPool(Key)]) \cdot \\
&\qquad Query \cdot (f^{conv}[MaxPool(Query); AvgPool(Query)])))
\end{aligned}
\tag{13}
$$

where $f^{Attention}$ is the self-attention weight allocation, and $f^{conv}$ is the convolution operation.

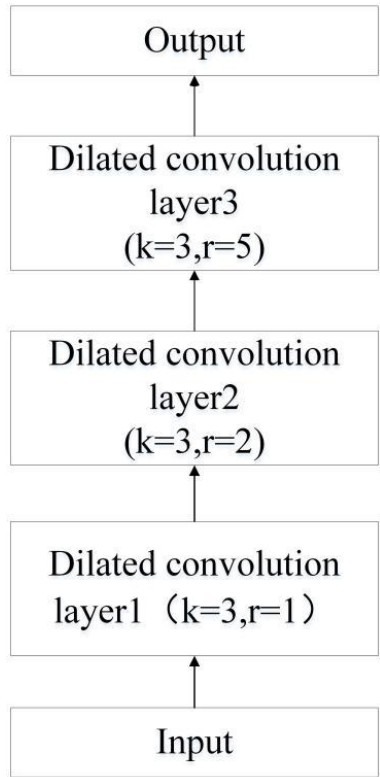

**Figure 8.** Dilated convolution block.

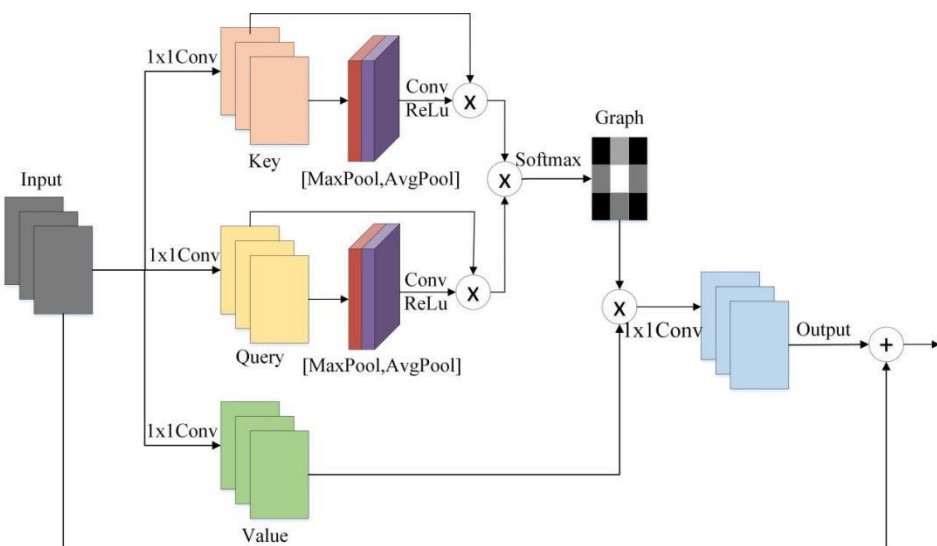

**Figure 9.** Self-attention mechanism with spatial self-attention.

The traditional convolution was replaced by dilated self-attention convolution as the input of the main capsule layer of the capsule network. The primary capsule layer converts scalar neurons into a primary capsule with a dimension of 8 in vector form. The spatial relationships among the local features extracted from the main capsule can be learned by the digital capsule layer, and fully connected ones are used to connect between the main capsule layer and the digital capsule layer and to assign weights though the dynamic routing algorithm (Algorithm 1). Dynamic routing is performed between two consecutive capsule layers to update the weight, and the assigned weight factors determine the transmission mode of the feature information from the low-level capsule to the high-level capsule.

The digital capsule layer transports the features to the fully connected layer, where ReLu is used as the activation function for the fully connected layer. The output of the fully connected layer can be determined by

$$z(t) = \left[z_1^t, z_2^t, z_3^t, \cdots, z_N^t\right] \tag{14}$$

---

**Algorithm 1.** Dynamic Routing Algorithm [21]

---

1  **procedure** ROUNTING($u_{j|i}$,,)
2 Initialize the coupling coefficients: $b_{ij} \leftarrow 0$
3  **for** $r$ iterations **do**
4    for all capsule $i$ in layer $l$: $c_{ij} \leftarrow \frac{e^{b_{ij}}}{\sum_j e^{b_{ij}}}$
5      for all capsule $j$ in layer $l+1$: $S_j \leftarrow c_{ij}u_{j|i}$
6      for all capsule $j$ in layer $l+1$: $V_j \leftarrow \frac{\|S_j\|^2}{1+\|S_j\|^2}\frac{S_j}{\|S_j\|}$
7      for all $b_{ij}$: $b_{ij} \leftarrow b_{ij} + u_{j|i}V_j$
8  **return** $V_j$

---

Temporal Features Captured by BiLSTM

The BiLSTM model is a composite of two directional LSTMs, which can realize bidirectional data processing of two separate hidden layers to be merged into the same output layer. The hidden layer states of the two LSTMs jointly determine the output results, which were used to extract the long-term correlation of the sequence data.

The current moment input is represented by

$$x(t) = \left[x_1^t, x_2^t, x_3^t, \cdots, x_M^t\right] \tag{15}$$

where $M$ is the two-dimensional matrix after dimensional change of the input vibration signal. The output feature of the LSTM at moment $t-1$ is shown as

$$h(t-1) = \left[h_1^{t-1}, h_2^{t-1}, h_3^{t-1}, \cdots, h_N^{t-1}\right] \tag{16}$$

where $N$ denotes the quantity of output feature maps.

The new candidate data $C_t$ was calculated for the information passed in through the input and the forgetting gate as

$$C_t = f(t)C_{t-1} + i(t)\tanh\left(\sum_{m=1}^{M} w_{m\varphi}x_m^t + b_{m\varphi} + \sum_{n=1}^{N} w_{n\varphi}h_n^{t-1} + b_{n\varphi}\right) \tag{17}$$

$$\tanh = \frac{e^x - e^{-x}}{e^x + e^{-x}} \tag{18}$$

where $f(t)$ is the output of the forget gate and $i(t)$ is the output of the input gate; the tanh function is a hyperbolic tangent function that normalizes the variables between $[-1,1]$. The $w_{m\varphi}$ and $b_{n\varphi}$ denote the weights and biases corresponding to the new information candidates $C_t$ of the $m$th feature map at the current moment, respectively; $w_{n\varphi}$ and $b_{n\varphi}$

denotes the weights and biases corresponding to the *n*th feature map at the recent moment, respectively.

The output gate $O(t)$ is designed to filter the cell state during the present moment.

$$O(t) = \sigma \left[ \sum_{m=1}^{M} w_{mr} x_m^t + b_{mr} + \sum_{n=1}^{N} w_{nr} h_n^{t-1} + b_{nr} \right] \tag{19}$$

where $w_{mr}$ and $b_{mr}$ denote the weights and biases corresponding to the output gates $O(t)$ of the *m*th feature map at the current moment, respectively, and $w_{nr}$ and $b_{nr}$ denote the weights and biases corresponding to the *n*th feature map at the previous moment, respectively.

The output $h(t)$ of the LSTM at the current moment is represented as

$$h(t) = O(t)\tanh(C_t) \tag{20}$$

### 3.3. ReLu Activation Function

The convolution is a linear operation, which is difficult to use to describe the complex relationship between signal and noise; therefore, activation functions are used to increase the nonlinear learning capability. The ReLu function was applied to activate the output of the convolution operation in a nonlinear manner, which has the advantages of preventing gradient scattering and sparsity and speeding up the computation, and can be expressed by

$$ReLu(x) = \max(0, x) = \begin{cases} x, x \geq 0 \\ 0, x < 0 \end{cases} \tag{21}$$

### 3.4. Performance Evaluation

The DACapsNet–BiLSTM network learns in a supervised manner to make predictions from noisy inputs to the original noiseless outputs, where minimization error estimates are used to reduce the errors present in the predictions.

The mean squared error was chosen as the training loss function, which can expressed by

$$\text{MSE} = \frac{1}{n} \sum_{i=1}^{n} \left( y_i' - y_i'' \right)^2 \tag{22}$$

where $y'_i$ represents the predicted output value and $y_i''$ represents the actual output value.

## 4. Experimental Results

### 4.1. Experimental Environment

The DACapsNet–BiLSTM model was trained using the deep learning module of Tensorflow 1.4.0 (San Francisco, USA) in an Intel(R) Core(TM) i5-9400F CPU@2.90GHz with 16 GB of RAM, and NVIDIA GeForce GTX 1080.The software programming environment is Python 3.6, and Spyder of Anaconda software 4.2.1 (Austin, TX, USA).

### 4.2. Pre-Processing

The minimization of the denoising error was applied to adaptively adjust the model arguments in the network learning, where the noisy signal is used as input and the pure signal is fitted as output, as shown in Figure 10. It is essential to normalize the vibration signal to ensure that the inputs have similar scales and the gradient descent algorithm can converge quickly.

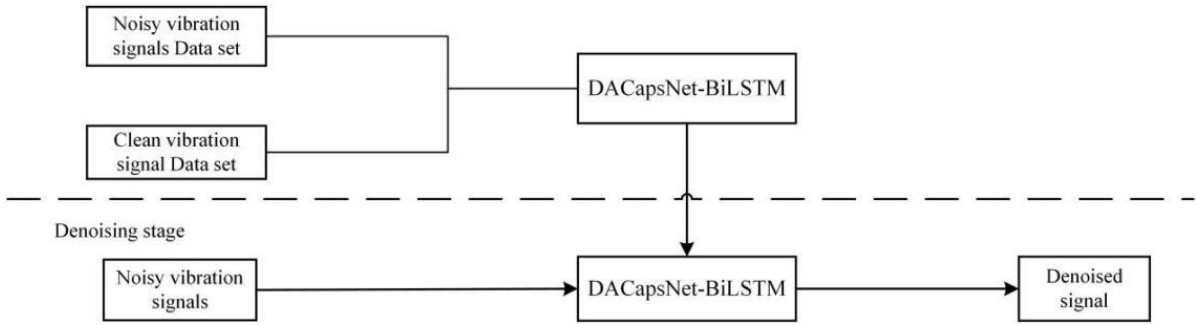

**Figure 10.** Network training and denoising process.

*4.3. Evaluation Indicators*

The following three metrics were used to evaluate the performance of the model proposed.

(1) Signal-to-noise ratio (SNR) was used to evaluate the signal output from the model in terms of the proportion of noise contained in the noise-containing signal by

$$\text{SNR} = 10 \times \log_{10} \frac{P_{signal}}{P_{noise}} = 10 \times \log_{10} \frac{\sum_1^n y_i^2}{\sum_1^n (y_i - y'_i)^2} \tag{23}$$

where $y_i$ is the noisy signal, $y'_i$ is the clean signal, and $n$ is the number of sampling points.

(2) The smaller value of mean square error (MSE) analysis of the error to evaluate the model, which reflects the degree of difference between the predicted values and the actual output. The MSE is calculated by

$$\text{MSE} = \frac{1}{n} \sum_{i=1}^n (y_i - y'_i)^2 \tag{24}$$

where $y_i$ is the actual output, $y'_i$ is the predicted values, and $n$ is the number of samples.

(3) Mean absolute error (MAE) is an effective error assessment method, which avoids errors canceling each other out by responding to the average of the absolute deviations of the predicted and true values. The MAE is calculated by

$$\text{MAE} = \frac{1}{n} \sum_{i=1}^n |y_i - y'| \tag{25}$$

where $y_i$ is the actual output, $y'_i$ is the predicted values, and $n$ is the number of samples.

*4.4. Simulation Experiment*

The fault model was used to simulate the impact signal caused by local defects in the bearing inner ring. White noise was added to the impact signal to simulate the early fault signal of the bearing inner race. The analog signal can be represented as

$$\begin{cases} x(t) = s(t) + n(t) = \sum_i A_i h(t - iT) + n(t) \\ h(t) = \exp(-Ct)\cos(2\pi f_n t) \\ A_i = 1 + A_0 \cos(2\pi f_r t) \end{cases} \tag{26}$$

where $s(t)$ is the cyclical impact component, $n(t)$ is the Gaussian white noise component. The SNR with noisy signal is $-5$ dB. $A_0$ is the amplitude, $f_r$ is the rotational frequency, $f_n$ is the resonance frequency and $C$ is the attenuation coefficient. $f_s$ is the sampling frequency of the analog signals.

The selection of these parameter values is shown in Table 1.

**Table 1.** The simulated signal parameter values.

| $A_0$ | $f_r$ | $C$ | $f_n$ | $f_s$ |
|---|---|---|---|---|
| 0.3 | 30 $Hz$ | 700 | 4 $KHz$ | 17 $KHz$ |

The total of 16,000,000 sample data points were collected to analyze the vibration signal analog signal. In order to maximize the feature analysis of the data, the collected sample points were reconstructed into 8000 samples, and each sample point has 2000 data points, where the training and test samples were divided in a 3:1 ratio. The $1 \times 2000$ one-dimensional data were transformed into a $50 \times 40$ two-dimensional image for processing, which reduces the complexity of signal handling and retains the periodic features of the input signals.

Gaussian white noise was used in the simulated signal as the noisy signal, in which the SNR of the noisy signal was $-10$ dB, $-5$ dB, 0 dB, 5 dB, and 10 dB. The simulated signal and the noisy signal were used as training samples, and their waveforms and spectra are shown in Figure 11.

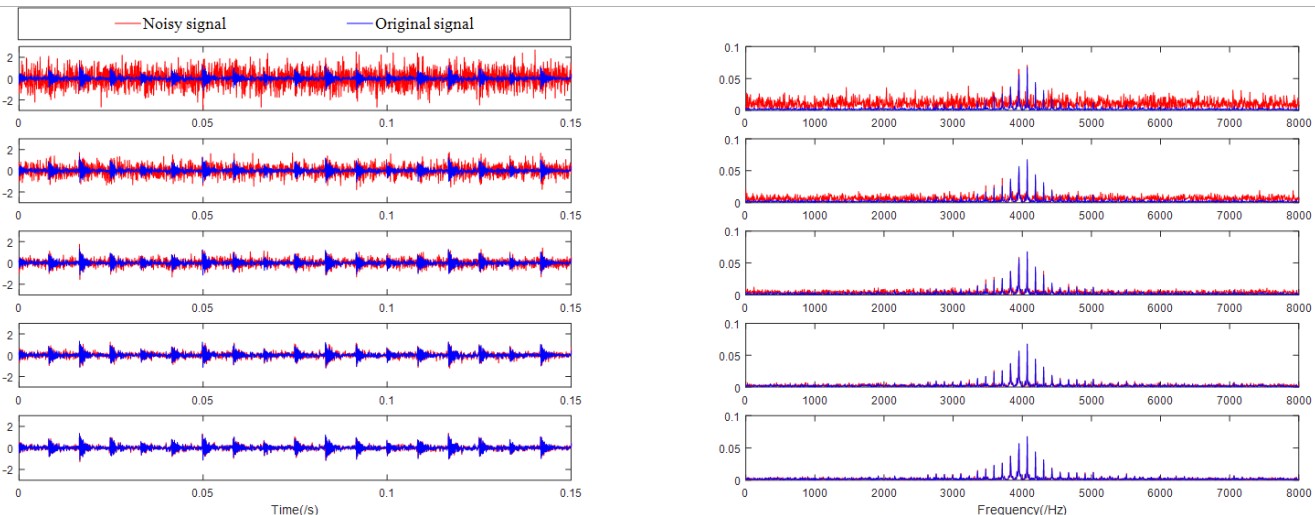

**Figure 11.** Time domain and spectrum of the noisy signal and the simulated signal (SNR = $-10$ dB, $-5$ dB, 0 dB, 5 dB and 10 dB).

The one-dimensional vibration measurement was converted into the two-dimensional vibration signal matrix, which were used as inputs to the DACapsNet–BiLSTM network. The improved capsule network is composed of two dilated self-attention convolution blocks, one primary capsule layer, one digital capsule layer, and one fully connected layer. In the dilated self-attentive convolutional block, three dilated convolutional layers are used to extract global features in the vibrating signal image and the ReLu activation function is used to perform nonlinear activation. The self-attention with location attention mechanism is embedded in the dilated self-attention convolution block. The output of the dilated self-attention convolution block is used as the input to the primary capsule layer, where the $5 \times 5$ convolution kernel is used to convolve to acquire a 32-way capsule feature map. The size of each feature map is $17 \times 12 \times 8$ and the weights of all capsules in the feature map are shared. The main capsule layer is fully connected to the digital capsule layer, and each weight is determined by a dynamic routing algorithm.

The digital capsule layer consists of 10 capsules, each with a size of $1 \times 16$. The processed features of the digital capsule layer are transported to the fully connected layer, which contains 2000 neurons. The temporal features of the vibration signal are extracted by a bidirectional long and short-term memory network, where the hidden layer cells are set to 256 and the $1 \times 2000$ feature map is used as the final output. The overall parameter design of the DACapsNet–BiLSTM is shown in Table 2.

**Table 2.** Parameter configuration of the DACapsNet–BiLSTM network model.

| | Parameters | | | | | Output Size |
|---|---|---|---|---|---|---|
| | Kernel Size | Channel | Rate | Stride | Capsule Dimension | |
| Input | / | / | / | / | / | $1 \times 2000$ |
| Reshape | / | / | / | / | / | $50 \times 40 \times 1$ |
| Conv 1 | $3 \times 3$ | 32 | 1 | 1 | / | $50 \times 40 \times 32$ |
| Conv 2 | $3 \times 3$ | 32 | 2 | 1 | / | $50 \times 40 \times 32$ |
| Conv 3 | $3 \times 3$ | 32 | 5 | 1 | / | $50 \times 40 \times 32$ |
| Improved Self-Attention1 | / | / | / | / | / | $50 \times 40 \times 32$ |
| Conv 4 | $3 \times 3$ | 32 | 1 | 1 | / | $50 \times 40 \times 32$ |
| Conv 5 | $3 \times 3$ | 32 | 2 | 1 | / | $50 \times 40 \times 32$ |
| Conv 6 | $3 \times 3$ | 32 | 5 | 1 | / | $50 \times 40 \times 32$ |
| Improved Self-Attention2 | / | / | / | / | / | $50 \times 40 \times 32$ |
| Primary capsule | $5 \times 5$ | 32 | / | 2 | 8 | $17 \times 12 \times 32 \times 8$ |
| Digital capsule | / | / | / | / | 16 | $10 \times 16$ |
| FC | / | / | / | / | / | $1 \times 2000$ |
| BiLSTM | / | / | / | / | / | $1 \times 2000$ |
| Output | / | / | / | / | / | $1 \times 2000$ |

The waveform, spectrogram and frequency plot of original vibration signal, the noise added vibration signal (SNR = −5 dB) and the denoised signal are shown as Figure 12. Figure 12 intuitively shows that most of the noise in the noisy signal after model learning has been eliminated.

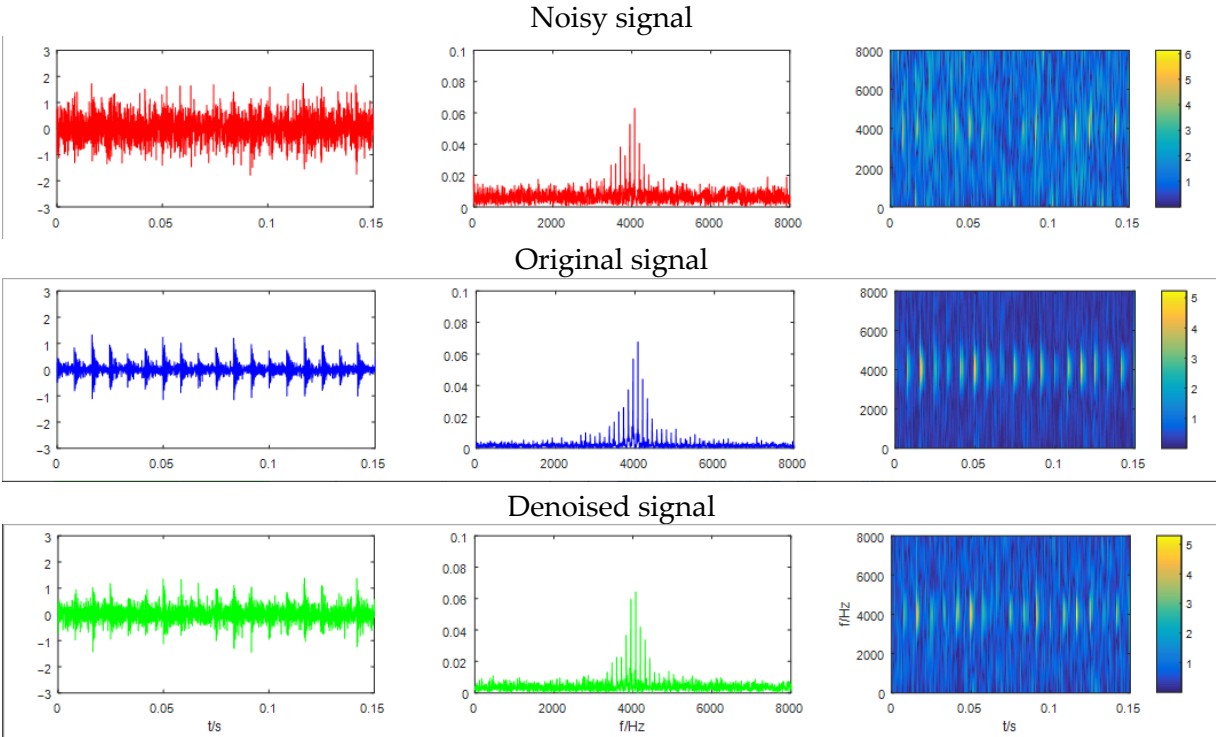

**Figure 12.** Simulation of signal waveforms, spectrograms and time-frequency diagrams.

In order to verify the robustness of the noise cancellation effect of the proposed method, Gaussian white noise of different intensities was mixed into the simulated signal to verify the robustness of the model (−10~10 dB). Three evaluation metrics, SNR, MSE and MAE, were applied to evaluation of the noise reduction performance of the model, and compared with other methods, including the wavelet threshold method (WT), empirical mode decomposition (EMD) method, CNN, Capsnet, LSTM, Vanilla LSTM, Stacked LSTM and BiLSTM. In addition to the above classical models, we have also added two recent deep learning models, BLC–CNN and ResNet–LSTM [41].

Among the above methods, WT is a classical signal-processing method, in which the choice of mother wavelet has an important influence on the performance of the WT

method. Usually, the wavelets used in the denoising process should satisfy the following properties, including orthogonality, symmetry, and regularity. The wavelet functions of the Daubechies family better reveal the periodic behavior of vibration signals [42]. The wavelet length of the mother wavelet function is a key determinant of signal denoising. Therefore, we selected wavelets of different lengths to verify the performance of the selected mother wavelet function under −10 SNR. The results are shown in Figure 13.

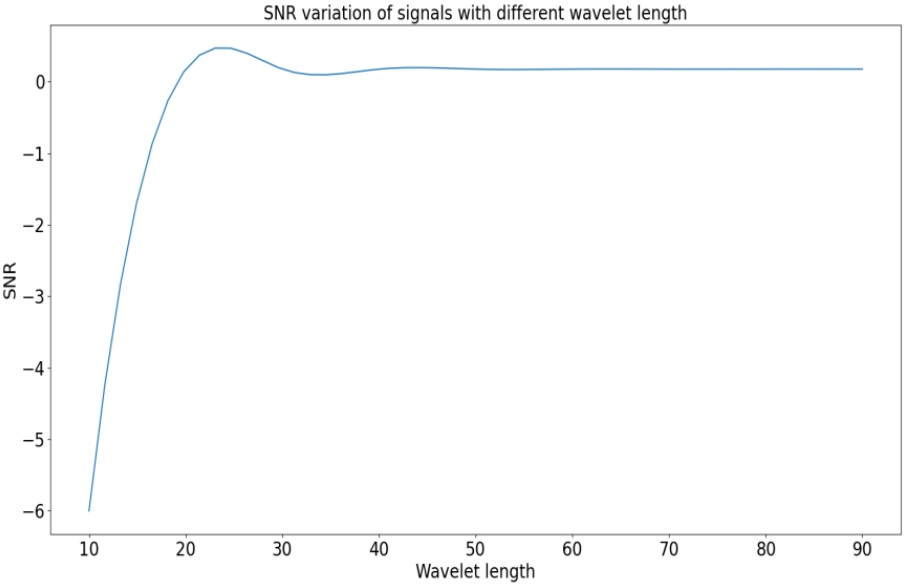

**Figure 13.** SNR variation of signals with different wavelet length.

As can be seen in Figure 6, the value of SNR increases with the increase of wavelet length. The value of SNR reaches the maximum when SNR equals to 24 (db12), and then the value of SNR remains stable after slightly decreasing with the increase of wavelet length, so db12 was used as the mother wavelet of the comparison model WT.

Table 3 shows the average SNR, MSE and MAE values of the test set of the traditional signal denoising method and the signal denoising method based on deep learning under the Gaussian white noise environment with five different SNRs. The results show that the signal quality after noise reduction using the DACapsNet–BiLSTM network model is significantly improved compared with other comparison methods. Specifically, when compared with the traditional methods, in most cases, the deep learning denoising method performed better in the three evaluation indexes. When compared with the CNN network, the denoising effect of the capsule network shows obvious advantages: the SNR is larger, and the evaluation indexes of Mae and MSE are more ideal. BiLSTM can further learn the influence of the information before and after each time point in the vibration data on the hidden features. When compared with LSTM, the denoising effect is better. When compared with the normal LSTM and superimposed LSTM, the DACapsNet–BiLSTM has lower MSE and MAE, so the denoising effect is better. The DACapsNet–BiLSTM has a significant performance when compared to the capsule network and BiLSTM network on three evaluation metrics. The proposed model has a higher denoising performance and better performance in various evaluation metrics by embedding extended self-focused convolution when compared to the BLC-CNN model. The performance of the proposed model is more convincing compared to the ResNet–LSTM model. In summary, DACapsNet–BiLSTM has excellent denoising performance and can achieve a better separation of noise and useful vibration signals, which is beneficial to the subsequent fault feature extraction.

**Table 3.** Comparison of noise reduction performance at different noise levels.

| Noise SNR (dB) | Evaluation Metrics | Original Signal | Traditional Method | | | | Deep Learning Methods | | | | | | |
|---|---|---|---|---|---|---|---|---|---|---|---|---|---|
| | | | WT | EMD | CNN | CapsNet | LSTM | Vanilla LSTM | Stacked LSTM | BiLSTM | BLC–CNN | ResNet–LSTM | DACapsNet–BiLSTM |
| −10 | SNR | −10.0008 | 0.1748 | −2.3971 | 0.2665 | 0.2783 | −0.0088 | 0.0021 | 0.0983 | 0.1024 | 0.3062 | 0.2794 | **0.3261** |
| | MSE | 0.5873 | 0.0564 | 0.1020 | 0.0552 | 0.0542 | 0.0615 | 0.0601 | 0.0562 | 0.0545 | 0.0537 | 0.0541 | **0.0532** |
| | MAE | 0.6116 | 0.1681 | 0.2410 | 0.1672 | 0.1655 | 0.1748 | 0.1732 | 0.1679 | 0.1659 | 0.1649 | 0.1652 | **0.1646** |
| −5 | SNR | −4.9975 | 1.3244 | 0.0161 | 1.0340 | 1.5813 | 1.4953 | 1.4836 | 1.8125 | 2.2774 | 2.7766 | 1.9263 | **3.2113** |
| | MSE | 0.1855 | 0.0432 | 0.0585 | 0.0482 | 0.0370 | 0.0395 | 0.0398 | 0.0364 | 0.0351 | 0.0335 | 0.0361 | **0.0329** |
| | MAE | 0.3436 | 0.1513 | 0.1645 | 0.1610 | 0.1439 | 0.1479 | 0.1481 | 0.1432 | 0.1427 | 0.1421 | 0.1438 | **0.1412** |
| 0 | SNR | −0.0008 | 2.8428 | 2.9396 | 3.3862 | 3.6590 | 3.4550 | 3.5283 | 3.8124 | 4.0892 | 4.8326 | 3.8714 | **5.2637** |
| | MSE | 0.0587 | 0.0305 | 0.0298 | 0.0286 | 0.0220 | 0.0256 | 0.0248 | 0.0201 | 0.0167 | 0.0163 | 0.0208 | **0.0162** |
| | MAE | 0.1933 | 0.1311 | 0.1301 | 0.1293 | 0.1184 | 0.1252 | 0.1236 | 0.1173 | 0.1134 | 0.1102 | 0.1161 | **0.1094** |
| 5 | SNR | 4.9978 | 6.9163 | 8.9529 | 9.4165 | 9.6525 | 9.4891 | 9.4893 | 9.7962 | 11.4734 | 11.9966 | 10.1771 | **12.6432** |
| | MSE | 0.0185 | 0.0119 | 0.0074 | 0.0068 | 0.0054 | 0.0058 | 0.0058 | 0.0053 | 0.0040 | 0.0037 | 0.0046 | **0.0032** |
| | MAE | 0.1087 | 0.0870 | 0.0688 | 0.0673 | 0.0649 | 0.0663 | 0.0663 | 0.0642 | 0.0621 | 0.0619 | 0.0641 | **0.0619** |
| 10 | SNR | 9.9906 | 11.3013 | 12.0532 | 13.4246 | 13.6537 | 13.4909 | 13.5238 | 13.5976 | 13.6255 | 14.4083 | 14.0801 | **14.4862** |
| | MSE | 0.0058 | 0.0039 | 0.0036 | 0.0033 | 0.0030 | 0.0032 | 0.0032 | 0.0031 | 0.0031 | 0.0029 | 0.0030 | **0.0029** |
| | MAE | 0.0611 | 0.0530 | 0.0477 | 0.0470 | 0.0467 | 0.0469 | 0.0468 | 0.0466 | 0.0467 | 0.0462 | 0.0464 | **0.0461** |

## 4.5. Experimental Analysis and Engineering Applications

In order to verify the effectiveness of the method for bearing fault diagnosis, the bearing vibration signal was denoised and analyzed.

Locomotive Bearing Vibration Signal Analysis

This experimental data comes from the vibration signal of the locomotive walking section gearbox. A locomotive travel section uses a rolling bearing model 552732QT, whose parameters are shown in Table 4. The top wheel test used a mobile top wheel device to jack up one side of the locomotive wheel pair and pick up the vibration signal when the wheel pair is rotating. The vibration acceleration sensor is installed in the upper position of the axle box. Figure 14 shows the test diagram of the top wheel of the locomotive. The speed of the axle was 515 r/min (8.58 Hz) and the sampling frequency was set to 12.8 kHz.

**Table 4.** Experimental rolling bearing parameters.

| Model | Inner Diameter | Outer Diameter | Roller Diameter | Number of Rollers | Contact Angle $\theta$ (Degree) |
|---|---|---|---|---|---|
| 552732QT | 160 (mm) | 290 (mm) | 34 (mm) | 17 | 0 |

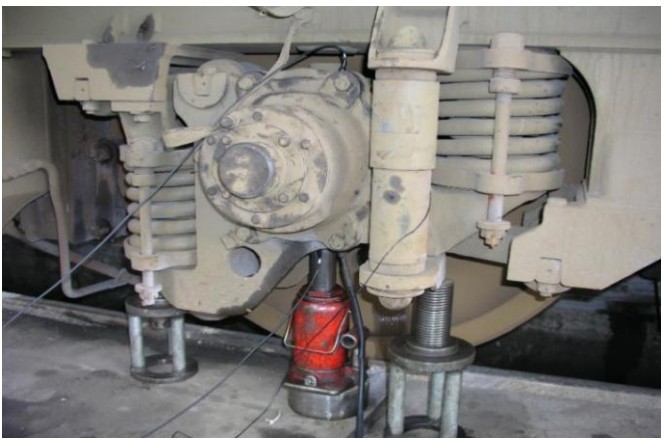

**Figure 14.** Locomotive rolling bearing top wheel test device.

This experimental data set contains a total of ten different health states of the bearing data, including six single scratch failure states, three compound scratch failure states and one health state, as shown in Figure 15. The waveforms of the bearing vibration data samples are shown, where the horizontal axis represents the sampling time and the vertical axis represents the amplitude. For locomotive bearing vibration signals, 16,000,000 sample data points were collected for data reconstruction.

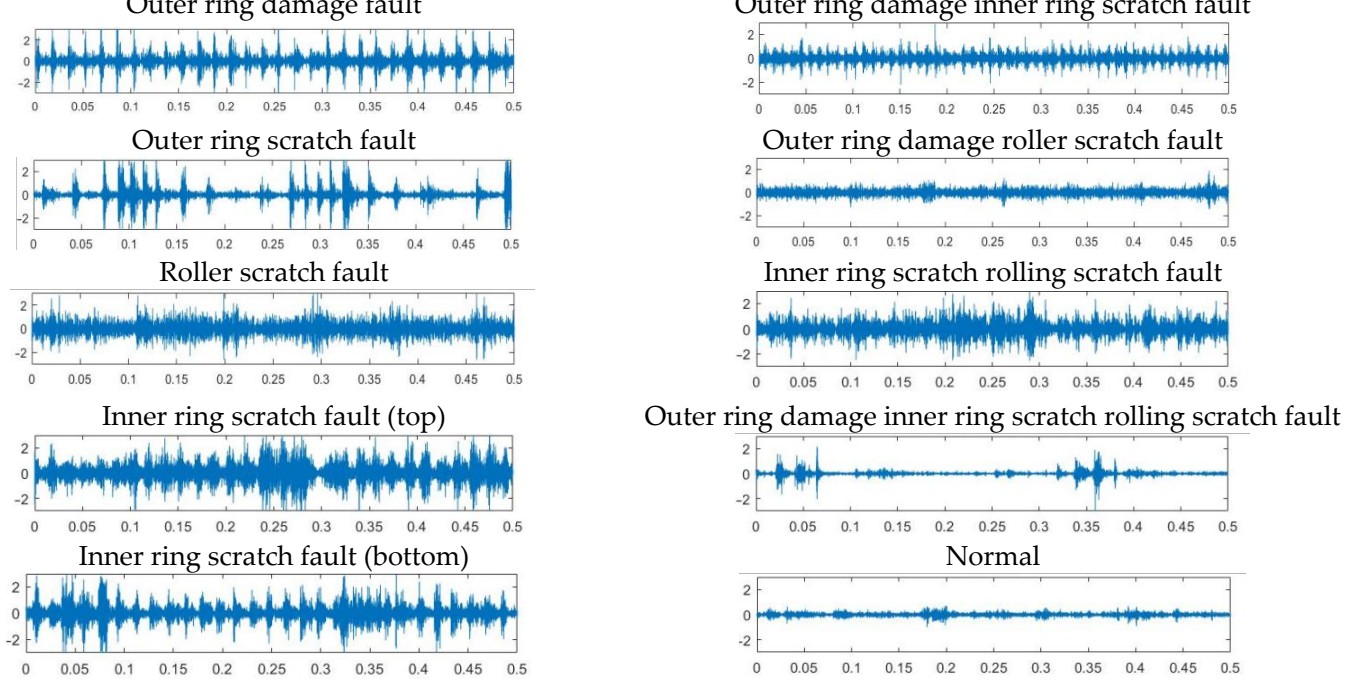

**Figure 15.** Time domain waveforms of bearing vibration signals for 10 states.

The Gaussian white noise with different signal-to-noise ratios was added to prove the performance of DACapsNet–BiLSTM network. The noise reduction effect of the model was evaluated using SNR, MSE, and MAE. The comparison of the proposed method with CNN, CapsNet, LSTM, Vanilla LSTM, Stacked LSTM, BiLSTM, BLC–CNN, and ResNet–CNN networks is shown as Table 5. Each model was experimented on using the same parameters as the previous model.

The results show that the DACapsNet–BiLSTM model works well at five different noise levels and outperforms other comparative methods. Specifically, although wavelet denoising methods and empirical mode decomposition methods can enhance the SNR of vibration signal to a limited degree, the improvement is limited due to complex data. The deep learning model can learn the noise in noisy signal through training methods, which can obtain the denoised signal with a higher SNR. Under different noise levels, compared with CapsNet and BiLSTM network models, the DACapsNet–BiLSTM model can obtain relatively high SNRs, especially when the SNR of noisy signals is negative. The higher SNR indicates that the DACapsNet–BiLSTM network has an excellent ability to remove noise, while the lower MSE and MAE indicators show that the DACapsNet–BiLSTM model not only has a higher noise reduction level, but also has less waveform distortion after noise reduction. The original vibration signal, the noisy signal (SNR = −5 dB), and the time domain waveforms of the denoised signal are shown in Figure 16. The noise information was removed in the time domain and the useful information is retained.

**Table 5.** Comparison of noise reduction performance at different noise levels.

| Noise SNR (dB) | Evaluation Metrics | Original Signal | Traditional Method | | Deep Learning Methods | | | | | | | | |
|---|---|---|---|---|---|---|---|---|---|---|---|---|---|
| | | | WT | EMD | CNN | CapsNet | LSTM | BiLSTM | Vanilla LSTM | Stacked LSTM | BLC–CNN | ResNet–LSTM | DACapsNet–BiLSTM |
| −10 | SNR | −9.9987 | 0.0384 | −1.6465 | 0.0946 | 0.2464 | −0.5444 | 0.1475 | 0.1511 | 0.2387 | 0.2639 | 0.2511 | **0.3092** |
| | MSE | 3.7104 | 0.3678 | 0.5422 | 0.3421 | 0.3311 | 0.3689 | 0.3358 | 0.3328 | 0.3298 | 0.3291 | 0.3302 | **0.3283** |
| | MAE | 1.5370 | 0.4074 | 0.5373 | 0.3933 | 0.3898 | 0.4052 | 0.3917 | 0.3913 | 0.3902 | 0.3899 | 0.3901 | **0.3892** |
| −5 | SNR | −4.9999 | 1.1645 | 1.9972 | 2.2323 | 2.3944 | 2.1180 | 2.2965 | 2.2864 | 2.4362 | 2.5466 | 2.4133 | **3.0625** |
| | MSE | 1.1737 | 0.2838 | 0.2343 | 0.2324 | 0.2256 | 0.2330 | 0.2303 | 0.2311 | 0.2247 | 0.2246 | 0.2252 | **0.2238** |
| | MAE | 0.8644 | 0.3978 | 0.3862 | 0.3840 | 0.3799 | 0.3851 | 0.3796 | 0.3804 | 0.3741 | 0.3713 | 0.3762 | **0.3635** |
| 0 | SNR | −0.0012 | 2.5660 | 2.6498 | 3.2902 | 3.4103 | 3.3522 | 3.8473 | 3.6176 | 3.8735 | 4.3613 | 3.7358 | **5.1271** |
| | MSE | 0.3712 | 0.2055 | 0.1958 | 0.1797 | 0.1743 | 0.1749 | 0.1724 | 0.1733 | 0.1698 | 0.1659 | 0.1711 | **0.1515** |
| | MAE | 0.4862 | 0.3266 | 0.3249 | 0.3228 | 0.3195 | 0.3206 | 0.3182 | 0.3197 | 0.3171 | 0.3169 | 0.3183 | **0.3162** |
| 5 | SNR | 5.0004 | 7.3964 | 8.5021 | 9.4298 | 9.6120 | 9.4749 | 10.0173 | 10.0972 | 10.5764 | 11.4395 | 10.4284 | **12.2110** |
| | MSE | 0.1173 | 0.0675 | 0.0524 | 0.0454 | 0.0411 | 0.0448 | 0.0378 | 0.0369 | 0.0355 | 0.0342 | 0.0388 | **0.0328** |
| | MAE | 0.2733 | 0.2031 | 0.1826 | 0.1779 | 0.1728 | 0.1753 | 0.1703 | 0.1701 | 0.1693 | 0.1686 | 0.1697 | **0.1682** |
| 10 | SNR | 9.9976 | 12.6933 | 12.9967 | 13.0822 | 13.2107 | 13.5083 | 14.0711 | 14.0773 | 14.0791 | 14.2370 | 14.2237 | **14.3027** |
| | MSE | 0.0371 | 0.0199 | 0.0186 | 0.0178 | 0.0173 | 0.0160 | 0.0147 | 0.0147 | 0.0146 | 0.0145 | 0.0146 | **0.0145** |
| | MAE | 0.1537 | 0.1101 | 0.1088 | 0.1072 | 0.1065 | 0.1061 | 0.1053 | 0.1052 | 0.1052 | 0.1049 | 0.1051 | **0.1047** |

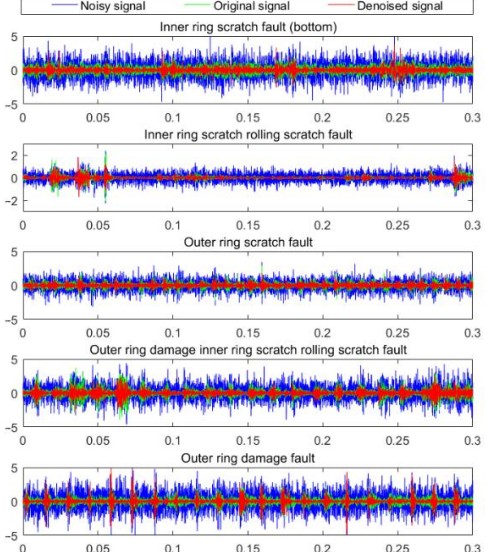
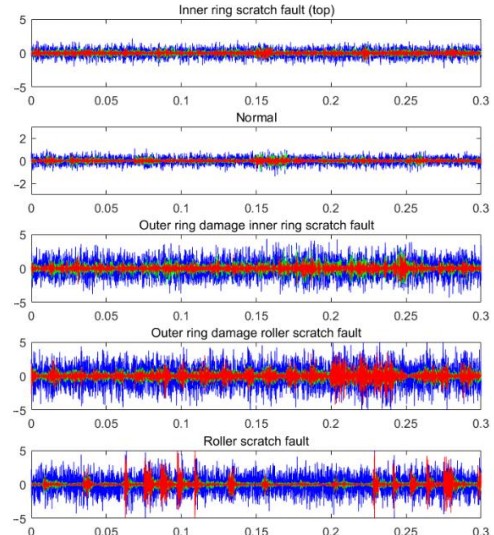

**Figure 16.** Vibration signals in 10 different states.

## 5. Conclusions

The convolutional neural network will lose a large amount of information when extracting the spatial features of vibration signals. An improved capsule neural network model was used to map the bearing vibration signal features to high-dimensional spatial features for learning. The dilated self-attentive convolutional block is integrated into the model to expand the receptive field, where an improved self-attentive module with positional attention was introduced to denoise the noise according to its random distribution. Considering the temporal features of the bearing vibration data, a BiLSTM network was added to extract the rich temporal information and achieve vibration signal denoising. The model effectively avoids the manual selection and fine-tuning of parameters by adaptively training parameters while learning nonlinear relationships between data. The DACapsNet–BiLSTM model outperforms other existing models in different evaluation metrics, and its performance in the SNR index has a significant advantage. As shown

above, DACapsNet–BiLSTM has improved performance and robustness compared with the prior denoising model.

Due to the fact that there is not enough fault data available for model training in industrial applications, the accuracy of the proposed models with small sample training is more demanding. Transfer learning can transfer the training data features of the relevant task to the target task, which provides us with new ideas for subsequent improvements. In future improvements, we aim to apply the knowledge of transfer learning to the proposed model to improve the robustness of our network under small samples and achieve satisfactory results in practical industrial applications.

**Author Contributions:** Conceptualization, Y.W. and G.C.; methodology, Y.W.; software, Y.W.; validation, Y.W., G.C. and J.H.; formal analysis, G.C.; investigation, G.C.; resources, Y.W.; data curation, Y.W.; writing—original draft preparation, Y.W.; writing—review and editing, Y.W.; visualization, Y.W.; supervision, Y.W.; project administration, Y.W.; funding acquisition, Y.W. All authors have read and agreed to the published version of the manuscript.

**Funding:** This work was supported by National Natural Science Foundation of China, approval number: 51875457; the Key Research and Development Program of Shanxi Province of China, approval number: 2022SF-259; the graduate student innovation fund of Xi'an University of Post and Telecommunications, approval number: CXJJLY202043.

**Data Availability Statement:** The data are available from the corresponding author upon reasonable request.

**Conflicts of Interest:** The authors declare no conflict of interest.

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
