# Peer review of "A Combination of Dilated Self-Attention Capsule Networks and Bidirectional Long- and Short-Term Memory Networks for Vibration Signal Denoising"

_machines, doi:10.3390/machines10100840_

Round 1

Reviewer 1 Report

1. Information lines 103-140 require confirmation for literary sources.

2. Line 150 you must provide the decoding of the character N.

3. Line 154. Figure 1 does not carry scientific content. It is recommended to remove it.

4. Line 176 you must provide the decoding of the character T.

5. Line 521. 00

Reviewer 2 Report

Authors presented an interesting work based on integration of dilated capsule networks with self-attention mechanism and bidirectional long and short-term memory. After reading the manuscript,there are certain queries which needs to be justified by authors and it is expected that manuscript is modified accordingly.

1. In pg.3,line 123-124,it is mentioned that A hybrid neural network for signal denoising is proposed. Why NN is needed for denoising signal when other signal processing methods like Wavelet Transform,Hilbert transform,EMD etc. are reported by various authors.

2. It is a good practice to compare the results from deep learning models. Authors reported the result of BiLSTM model.It is expected that results should also be included from other LSTM models like Stacked LSTM,Vanilla LSTM etc. so that comparative analysis can be done.

3. In pg.13, line 411-412,statement regarding MSE is not clear.Kindly re-write it.

4. In table 3, Wavelet Transform and EMD is mentioned as a traditional method for denoising. There are several mother wavelets available.Which wavelet authors selected for denoising and what is the criteria. Author should refer to recently published literature :

a. https://www.mdpi.com/2075-1702/10/3/176

b. https://www.mdpi.com/1424-8220/21/5/1851

Kindly add the relevant discussion and add justification in revised manuscript with addition of suitable literatures.

5. If possible, title can be reframed considering the inclusion of relevant keywords.

6. It is suggested to include recently published literature with suitable discussion to justify the utility of author's proposed methodologies.

7. Kindly add the limitations and future scope of methodologies proposed by authors.

Round 2

Reviewer 2 Report

Authors addressed reviewer comments and revised manuscript accordingly.